# Learning nonlinear operators in latent spaces for real-time predictions of complex dynamics in physical systems

Katiana Kontolati[1,3], Somdatta Goswami[2,3], George Em Karniadakis [2] & Michael D. Shields [1] ✉

Predicting complex dynamics in physical applications governed by partial differential equations in real-time is nearly impossible with traditional numerical simulations due to high computational cost. Neural operators offer a solution by approximating mappings between infinite-dimensional Banach spaces, yet their performance degrades with system size and complexity. We propose an approach for learning neural operators in latent spaces, facilitating real-time predictions for highly nonlinear and multiscale systems on high-dimensional domains. Our method utilizes the deep operator network architecture on a low-dimensional latent space to efficiently approximate underlying operators. Demonstrations on material fracture, fluid flow prediction, and climate modeling highlight superior prediction accuracy and computational efficiency compared to existing methods. Notably, our approach enables approximating large-scale atmospheric flows with millions of degrees, enhancing weather and climate forecasts. Here we show that the proposed approach enables real-time predictions that can facilitate decision-making for a wide range of applications in science and engineering.

Achieving universal function approximation is one of the most important tasks in the rapidly growing field of machine learning (ML). To this end, deep neural networks (DNNs) have been actively developed, enhanced, and used for a plethora of versatile applications in science and engineering including image processing, natural language processing (NLP), recommendation systems, and design optimization[1–6]. In the emerging field of scientific machine learning (SciML), DNNs are a ubiquitous tool for analyzing, solving, and optimizing complex physical systems modeled with partial differential equations (PDEs) across a range of scenarios, including different initial and boundary conditions (ICs, BCs), model parameters and geometric domains. Such models are trained from a finite dataset of labeled observations generated from a (generally expensive) traditional numerical solver (e.g., finite difference method (FD), finite elements (FEM), computational fluid dynamics (CFD), and once trained they allow for accurate predictions with real-time inference[7–10]).

DNNs are conventionally used to learn functions by approximating mappings between finite dimensional vector spaces. Operator regression, a more recently proposed ML paradigm, focuses on learning operators by approximating mappings between abstract infinite-dimensional Banach spaces. Neural operators specifically, first introduced in 2019 with the deep operator network (DeepONet)[11], employ DNNs to learn PDE operators and construct a surrogate model, which allows for fast inference and high generalization accuracy. Motivated by the universal approximation theorem for operators proposed by Chen & Chen[12], DeepONet encapsulates and extends the theorem for deep neural networks[11]. The architecture of DeepONet features a DNN, which encodes the input functions at fixed sensor points (branch net), and another DNN, which encodes the information related to the spatio-temporal coordinates of the output function (trunk net). Since its first appearance, standard DeepONet has been employed to tackle challenging problems involving complex

[1]Department of Civil and Systems Engineering, Johns Hopkins University, Baltimore, ML 21218, USA. [2]Division of Applied Mathematics, Brown University, Providence, RI 2906, USA. [3]These authors contributed equally: Katiana Kontolati, Somdatta Goswami. ✉e-mail: michael.shields@jhu.edu

high-dimensional dynamical systems[13–17]. In addition, extensions of DeepONet have been recently proposed in the context of multi-fidelity learning[18–20], integration of multiple-input continuous operators[21,22], hybrid transferable numerical solvers[23], transfer learning[24], and physics-informed learning to satisfy the underlying PDE[25,26].

Another class of neural operators is the integral operators, first instantiated with the graph kernel networks (GKN) introduced by[27]. In GKNs, the solution operator is expressed as an integral operator of Green's function which is modeled with a neural net and consists of a lifting layer, iterative kernel integration layers, and a projection layer. GKNs were found to be unstable for multiple layers and a new graph neural operator was developed in[28] based on a discrete non-local diffusion-reaction equation. Furthermore, to alleviate the inefficiency and cost of evaluating integral operators, the Fourier neural operator (FNO)[29] was proposed, in which the integral kernel is parameterized directly in the Fourier space. The input to the network, like in GKNs, is elevated to a higher dimension and then passed through numerous Fourier layers before being projected back to the original dimension. Each Fourier layer involves a forward fast Fourier transform (FFT), followed by a linear transformation of the low-Fourier modes and then an inverse FFT. Finally, the output is added to a weight matrix, and the sum is passed through an activation function to introduce nonlinearity. Different variants of FNO have been proposed, such as the FNO-2D which performs 2D Fourier convolutions and uses a recurrent structure to propagate the PDE solution in time, and the FNO-3D, which performs 3D Fourier convolutions through space and time. Compared to DeepONet, FNO, in its seminal paper[29], employs evaluations restricted to an equispaced mesh to discretize both the input and output spaces, where the mesh and the domain must be the same. The interested reader is referred to[30] for a comprehensive comparison between DeepONet and FNO across a range of complex applications. Recent advancements in neural operator research have yielded promising results for addressing the bottleneck of FNO. Two such integral operators are the Wavelet Neural Operator (WNO)[31] and the Laplace Neural Operator (LNO)[32], which have been proposed as alternative solutions for capturing the spatial behavior of a signal and accurately approximating transient responses, respectively.

Despite the impressive capabilities of the aforementioned methods to learn surrogates for complex PDEs, these models are primarily used in a data-driven manner, and thus a representative and sufficient labeled dataset needs to be acquired a-priori. Often, complex physical systems require high-fidelity simulations defined on fine spatial and temporal grids, which results in very high-dimensional datasets. Furthermore, the high (and often prohibitive) expense of traditional numerical simulators e.g., FEM allows for the generation of only a few hundred (and possibly even fewer) observations. The combination of few and very high-dimensional observations can result in sparse datasets that often do not represent adequately the input/output distribution space. In addition, raw high-dimensional physics-based data often consists of redundant features that can (often significantly) delay and hinder network optimization. Physical constraints cause the data to live on lower-dimensional latent spaces (manifolds) that can be identified with suitable linear or nonlinear dimension reduction (DR) techniques. Previous studies have shown how latent representations can be leveraged to enable surrogate modeling and uncertainty quantification (UQ) by addressing the 'curse of dimensionality' in high-dimensional PDEs with traditional approaches such as Gaussian processes (GPs) and polynomial chaos expansion (PCE)[33–37]. Although neural network-based models can naturally handle high-dimensional input and output datasets, it is not clear how their predictive accuracy, generalizability, and robustness to noise are affected when these models are trained with suitable latent representations of the high-dimensional data.

In this work, we aim to investigate the aforementioned open questions by exploring the training of DeepONet on latent spaces for high-dimensional time-dependent PDEs of varying degrees of complexity. The idea of training neural operators on latent spaces using DeepONet and autoencoders (AE) was originally proposed in[16]. In this work, the growth of a two-phase microstructure for particle vapor deposition was modeled using the Cahn-Hilliard equation. In another recent work[38], the authors explored neural operators in conjunction with AE to tackle high-dimensional stochastic problems. But the general questions of the predictive accuracy and generalizability of DeepONet trained on latent spaces remain and require systematic investigation with comparisons to conventional neural operators.

The training of neural operators on latent spaces consists of a two-step approach: first, training a suitable AE model to identify a latent representation for the high-dimensional PDE inputs and outputs, and second, training a DeepONet model and employing the pre-trained AE decoder to project samples back to the physically interpretable high-dimensional space (see Fig. 1). Related methods, in particular, the U-Net framework within the U-shaped neural operator (U-NO)[39] have aimed to achieve a similar objective. However, it's important to note that, while the U-Net framework within U-NO is commonly recognized as having encoder and decoder segments, these segments do not act as independent encoder and decoder components. Therefore, unlike AEs (and many other unsupervised dimension reduction methods), the encoder and decoder components cannot be disentangled from the original, high-dimensional data. The benefit of the proposed L-DeepONet framework, on the other hand, is that it is designed with independent encoder and decoder components to allow the direct construction of a neural operator in an arbitrarily learned low-dimensional space. It is therefore not constrained by the architecture or design of the encoder/decoder, which may be an AE (as studied here) or a different unsupervised dimension reduction method altogether[37].

The L-DeepONet framework has two advantages: first, the accuracy of DeepONet is improved, and second, the L-DeepONet training is accelerated due to the low dimensionality of the data in the latent space. Combined with the pre-trained AE model, L-DeepONet can perform accurate predictions with real-time inference and learn the solution operator of complex time-dependent PDEs in low-dimensional space. The contributions of this work can be summarized as follows:

- We investigate the performance of L-DeepONet, an extension of standard DeepONet, for high-dimensional time-dependent PDEs that leverages latent representations of input and output functions identified by suitable autoencoders (see Fig. 1).
- We perform direct comparisons with vanilla DeepONet for complex physical systems, including brittle fracture of materials, and complex convective and atmospheric flows, and demonstrate that L-DeepONet consistently outperforms the standard approach in terms of accuracy and computational time.
- We perform direct comparisons with another neural operator model, the Fourier neural operator (FNO), and two of its variants, i.e., FNO-2D and FNO-3D, and identify advantages and limitations for a diverse set of applications.
- We perform direct comparisons with U-NO and report the accuracy, computational time, and the number of trainable parameters in the Supplementary Tables S2 and S3.

For all the problems considered in this work, we have generated the training data with a fixed spatio-temporal discretization to deploy an AE for dimensionality reduction. However, for more general problems where training data vary in fidelity or where the training data is provided at arbitrary points in space and time, linear (e.g. principal component analysis, linear discriminant analysis[40]) and non-linear projection (e.g. bicubic interpolation, t-SNE[41], diffusion maps[42]) methods can be employed to create a shared continuous basis onto which

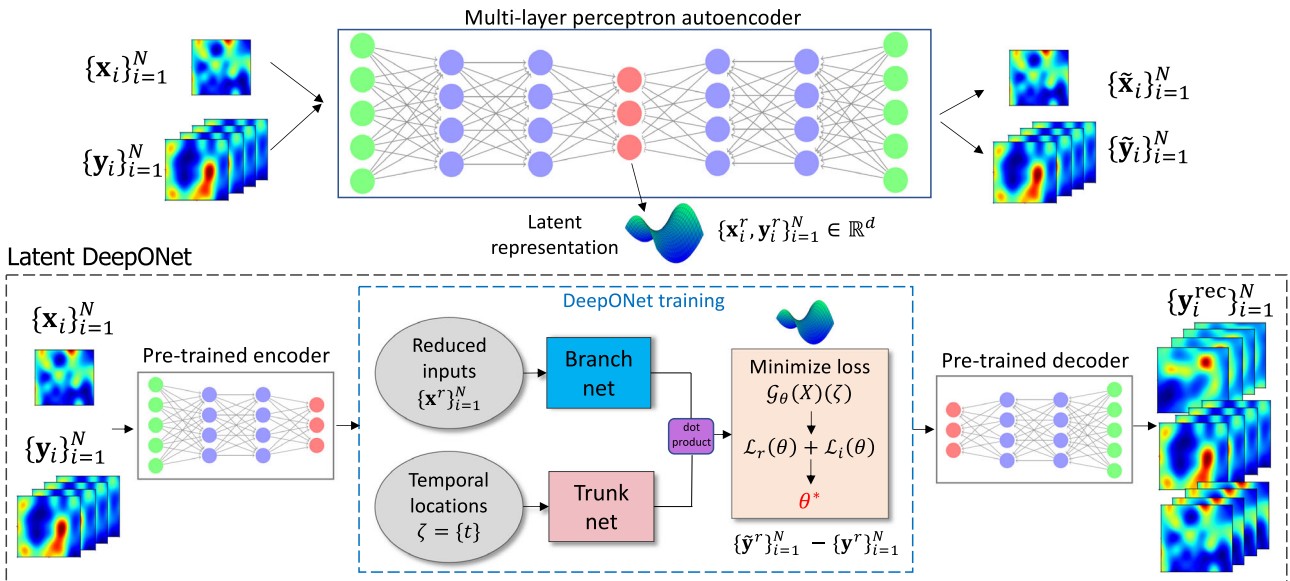

**Fig. 1 | Latent DeepONet (L-DeepONet) framework for learning deep neural operators on latent spaces.** In the first step, a multi-layer autoencoder is trained using a combined dataset of the high-dimensional input and output realizations of a PDE model, $\{\mathbf{x}_i, \mathbf{y}_i\}_{i=1}^N$, respectively. The trained encoder projects the data onto a latent space $\mathbb{R}^d$ and the dataset on the latent space, $\{\mathbf{x}_i^r, \mathbf{y}_i^r\}_{i=1}^N$ is then used to train a

DeepONet model and learn the operator $\mathscr{G}_\theta$, where $\theta$ denotes the trainable parameters of the network. Finally, to evaluate the performance of the model on the original PDE outputs and perform inference, the pre-trained decoder is employed to map predicted samples back to physically interpretable space.

the training data can be projected. This dual functionality offers the option to directly reduce dimensionality to enable the training of the DeepONet directly from the projected data. Alternatively, it facilitates the interpolation of the given training data onto a fixed grid, aligning with the requirement of AE. However, the implementation of such methods is beyond the scope of this work.

## Results

To demonstrate the advantages and efficiency of L-DeepONet, we learn the operator for three diverse PDE models of increasing complexity and dimensionality. First, we consider a PDE that describes the growth of fracture in brittle materials which are widely used in various industries including construction and manufacturing. Predicting with accuracy the growth of fractures in these materials is important for preventing failures, improving safety, reliability, and cost-effectiveness in a wide range of applications. Second, we consider a PDE describing convective fluid flow, a common phenomenon in many natural and industrial processes. Understanding how these flows evolve may allow engineers to better design systems such as heat exchangers or cooling systems to enhance efficiency and reduce energy consumption. Finally, we consider a PDE describing large-scale atmospheric flows which can be used to predict patterns that occur in weather systems. Such flows play a crucial role in the Earth's climate system influencing precipitation, and temperature which in turn may have a significant impact on water resources, agricultural productivity, and energy production. Developing an accurate surrogate to predict with detail such complex atmospheric patterns may allow us to better adapt to changes in the climate system and develop effective strategies to mitigate the impacts of climate change. For all PDEs, the input functions for the operator represent initial conditions modeled as Gaussian or non-Gaussian random fields. We perform direct comparisons of L-DeepONet with the standard DeepONet model trained on the full dimensional data and with FNO. More details about the models and the corresponding data generation process are provided in the Supplementary Section on Data Generation to assist the readers in readily reproducing the results presented below.

## Brittle fracture in a plate loaded in shear

Fracture is one of the most commonly encountered failure modes in engineering materials and structures. Defects, once initialized, can lead to catastrophic failure without warning. Therefore, from a safety point of view, prediction of the initiation and propagation of cracks is of utmost importance. In the phase field fracture modeling approach, the effects associated with crack formation, such as stress release, are incorporated into the constitutive model[43]. Modeling fracture using the phase field method involves the integration of two fields, namely the vector-valued elastic field, $\boldsymbol{u}(\boldsymbol{x})$, and the scalar-valued phase field, $\phi(\boldsymbol{x}) \in [0,1]$, with 0 representing the undamaged state of the material and 1 a fully damaged state.

The equilibrium equation for the elastic field for an isotropic model, considering the evolution of crack, can be written as[44]:

$$-\nabla \cdot g(\phi)\boldsymbol{\sigma} = \boldsymbol{f} \text{ on } \Omega, \tag{1}$$

where $\boldsymbol{\sigma}$ is the Cauchy stress tensor, $\boldsymbol{f}$ is the body force and $g(\phi) = (1-\phi)^2$ represents the monotonically decreasing stress-degradation function that reduces the stiffness of the bulk material in the fracture zone. The elastic field is constrained by Dirichlet and Neumann boundary conditions:

$$g(\phi)\boldsymbol{\sigma} \cdot \boldsymbol{n} = \boldsymbol{t}_N \text{ on } \partial\Omega_N,$$
$$\boldsymbol{u} = \overline{\boldsymbol{u}} \text{ on } \partial\Omega_D, \tag{2}$$

where $\boldsymbol{t}_N$ is the prescribed boundary forces and $\overline{\boldsymbol{u}}$ is the prescribed displacement for each load step. The Dirichlet and Neumann boundaries are represented by $\partial\Omega_D$ and $\partial\Omega_N$, respectively. Considering the second-order phase field for a quasi-static setup, the governing equation can be written as:

$$\frac{G_c}{l_0}\phi - G_c l_0 \nabla^2 \phi = -g'(\phi)H(\boldsymbol{x}, t; l_c, y_c) \text{ on } \Omega, \tag{3}$$

where $G_c$ is a scalar parameter representing the critical energy release rate of the material, $l_0$ is the length scale parameter, which controls the

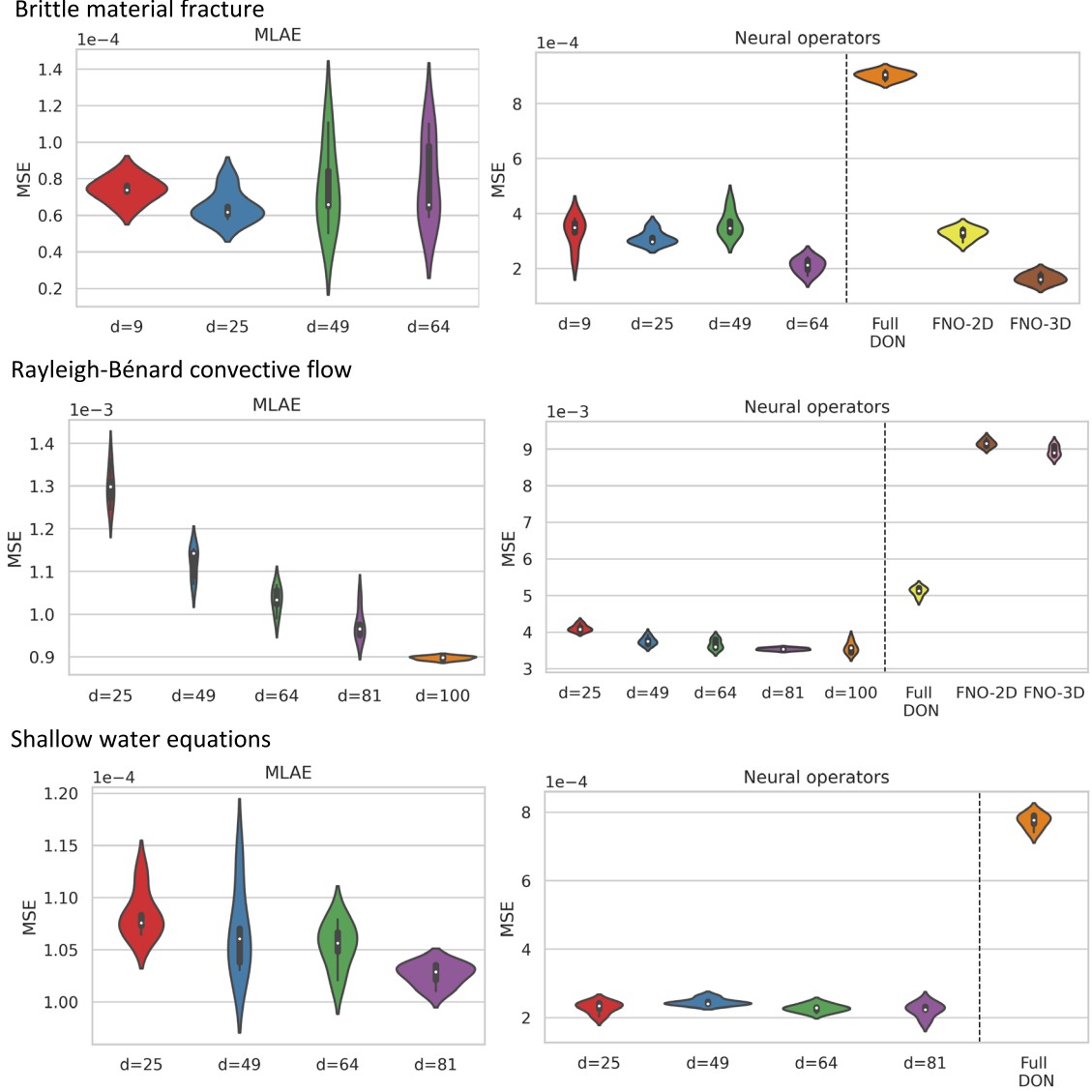

**Fig. 2 | Results of all applications.** Left: Results for the multi-layer autoencoders (MLAE) for different values of the latent dimensionality. Right: Results for all the studied neural operators. For all panes, violin plots are generated from 5 independent trainings of the models using different random seed numbers.

diffusion of the crack, $H(\mathbf{x}, t)$ is a local strain-history functional, and $y_c$, $l_c$ represent the position and length of the crack respectively. For sharp crack topology, $l_0 \to 0^{45}$. $H(\mathbf{x}, t)$ contains the maximum positive tensile energy ($\Psi_0^+$) in the history of deformation of the system. The strain-history functional is employed to initialize the crack on the domain as well as to impose irreversibility conditions on the crack growth[46]. In this problem, we consider $y_c$, $l_c$ to be random variables with $y_c \sim U[0.3, 0.7]$ and $l_c \sim U[0.4, 0.6]$, thus, the initial strain function $H(\mathbf{x}, t = 0; l_c, y_c)$ is also random (see the Supplementary Section on Data Generation). We aim to learn the solution operator $\mathcal{G} : H(\mathbf{x}, t = 0; l_c, y_c) \mapsto \phi(\mathbf{x}, t)$ which maps the initial strain-history function to the crack evolution.

In Fig. 2a, we show the mean-square error (MSE) between the studied models and ground truth. The left panel shows the MSE for the multi-layer autoencoder (MLAE) for different latent dimensions ($d$), where the violin plot shows the distribution of MSE from $n = 5$ independent trials. The right panel shows the resulting MSE for L-DeepONet operating on different latent dimensions ($d$) compared with the full high-dimensional DeepONet, FNO-2D, and FNO-3D. We observe that, regardless of the latent dimension, the L-DeepONet outperforms the standard DeepONet (Full DON) and performs comparably with FNO-2D

and FNO-3D. In Fig. 3, a comparison between all models for a random representative result is shown. While L-DeepONet results in prediction fields almost identical to the reference, the predictions of the standard models deviate from the ground truth both inside and around the propagated crack. Finally, the cost of training the different models is presented in Table 1. Because the required network complexity is significantly reduced, the L-DeepONet is $1 - 2$ orders of magnitude cheaper to train than the standard approaches.

**Rayleigh-Bénard fluid flow convection**
Rayleigh-Bénard convection occurs in a thin layer of fluid that is heated from below[47]. The natural fluid convection is buoyancy-driven and caused due to a temperature gradient $\Delta T$. Instability in the fluid occurs when $\Delta T$ is large enough to make the non-dimensional Rayleigh number, Ra, exceed a certain threshold. The Rayleigh number whose physical interpretation is the ratio between the buoyancy and the viscous forces is defined as

$$\mathrm{Ra} = \frac{\alpha \Delta T g h^3}{\nu \kappa}, \tag{4}$$

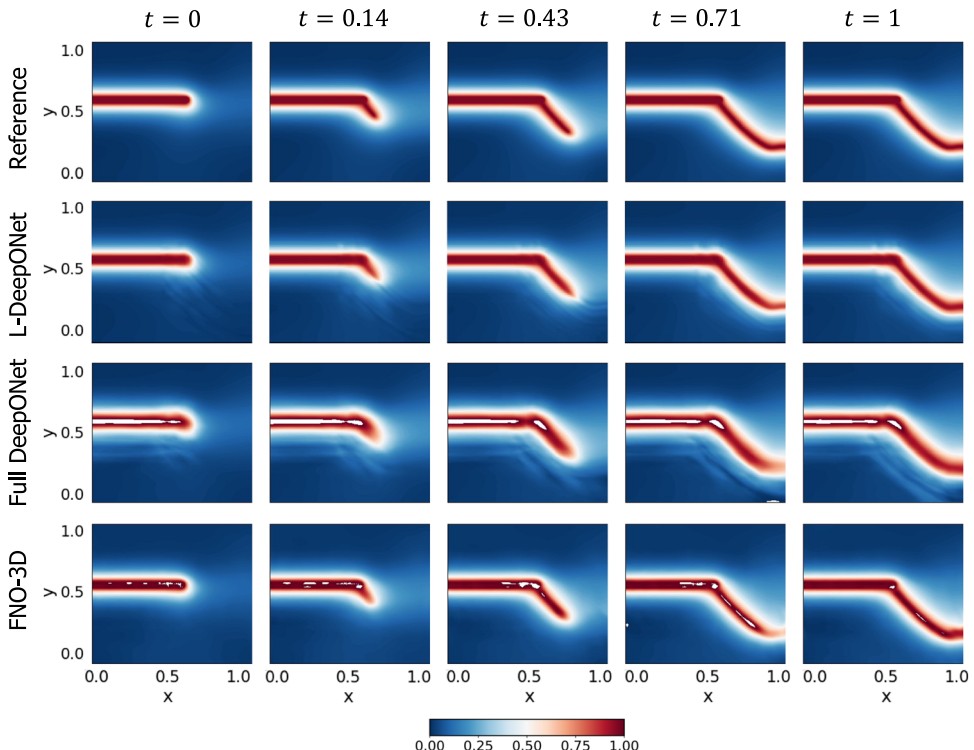

**Fig. 3 | Brittle fracture in a plate loaded in shear: results of a representative sample with $y_c = 0.55$ and $l_c = 0.6$ for all neural operators.** The results of the L-DeepONet model consider the latent dimension, $d = 64$. The neural operator is trained to approximate the growth of the crack for five time steps from a given initial location of the defect.

where $\alpha$ is the thermal expansion coefficient, $g$ is the gravitational acceleration, $h$ is the thickness of the fluid layer, $\nu$ is the kinematic viscosity and $\kappa$ is the thermal diffusivity. When $\Delta T$ is small, the convective flow does not occur due to the stabilizing effects of viscous friction. Based on the governing conservation laws for an incompressible fluid (mass, momentum, energy) and the Boussinesq approximation according to which density perturbations affect only the gravitational force, the dimensional form of the Rayleigh-Bénard equations for a fluid defined on a domain $\Omega$ reads:

$$\begin{cases} \frac{\mathrm{D}\boldsymbol{u}}{\mathrm{D}t} = -\frac{1}{\rho_0}\nabla p + \frac{\rho}{\rho_0}g + \nu\nabla^2\boldsymbol{u}, & \boldsymbol{x} \in \Omega, t > 0, \\ \frac{\mathrm{D}T}{\mathrm{D}t} = \kappa\nabla^2 T, & \boldsymbol{x} \in \Omega, t > 0, \\ \nabla \cdot \boldsymbol{u} = 0, \\ \rho = \rho_0(1 - \alpha(T - T_0)), \end{cases} \quad (5)$$

where $\mathrm{D}/\mathrm{D}t$ denotes material derivative, $\boldsymbol{u}, p, T$ are the fluid velocity, pressure and temperature respectively, $T_0$ is the temperature at the lower plate, and $\boldsymbol{x} = (x, y)$ are the spatial coordinates. Considering two plates (upper and lower) the corresponding BCs and ICs are defined as

$$\begin{cases} T(\boldsymbol{x},t)|_{y=0} = T_0, & \boldsymbol{x} \in \Omega, t > 0, \\ T(\boldsymbol{x},t)|_{y=h} = T_1, & \boldsymbol{x} \in \Omega, t > 0, \\ \boldsymbol{u}(\boldsymbol{x},t)|_{y=0} = \boldsymbol{u}(\boldsymbol{x},t)|_{y=h} = 0, & \boldsymbol{x} \in \Omega, t > 0, \\ T(y,t)|_{t=0} = T_0 + \frac{y}{h}(T_1 - T_0) + 0.1\upsilon(\boldsymbol{x}), & \boldsymbol{x} \in \Omega, \\ \boldsymbol{u}(\boldsymbol{x},t)|_{t=0} = 0, & \boldsymbol{x} \in \Omega, \end{cases} \quad (6)$$

where $T_0$, and $T_1$ are the fixed temperatures of the lower and upper plates, respectively. For a 2D rectangular domain and through a non-dimensionalization of the above equations, the fixed temperatures become $T_0 = 0$ and $T_1 = 1$. The IC of the temperature field is modeled as linearly distributed with the addition of a GRF, $\upsilon(\boldsymbol{x})$ having correlation length scales $\ell_x = 0.45, \ell_y = 0.4$ simulated using a Karhunen-Loéve

expansion. The objective is to approximate the operator $\mathscr{G}$: $T(\boldsymbol{x}, t = 0) \mapsto T(\boldsymbol{x}, t)$ (see the Supplementary Section on Data Generation).

Figure 2b again shows violin plots of the MSE for the MLAE with differing latent dimensions and the MLE for the corresponding L-DeepONet compared with the other neural operators. Here we see that the reconstruction accuracy of the MLAE is improved by increasing the latent dimensionality up to $d = 100$. However, the change in the predictive accuracy of L-DeepONet for different values of $d$ is less significant, indicating that latent spaces with even very small dimensions ($d = 25$) result in a very good performance. Furthermore, L-DeepONet outperforms all other neural operators with a particularly significant improvement compared to FNO. In Fig. 4, we observe that L-DeepONet is able to capture the complex dynamical features of the true model with high accuracy as the simulation evolves. In contrast, the standard DeepONet and FNO result in diminished performance as they tend to smooth out the complex features of the true temperature fields. Furthermore, the training time of the L-DeepONet is significantly lower than the full DeepONet and FNO as shown in Table 1.

### Shallow-water equations

The shallow-water equations model the dynamics of large-scale atmospheric flows[48]. In a vector form, the viscous shallow-water

**Table 1 | Comparison of the computational training time in seconds (s) for all the neural operators across all considered applications, identically trained on an NVIDIA A6000 GPU**

| Application | L-DeepONet | Full DeepONet | FNO-3D |
|---|---|---|---|
| Brittle material fracture | 1660 | 15,031 | 128,000 |
| Rayleigh-Bénard fluid flow | 2853 | 6772 | 1,126,400 |
| Shallow water equation | 15,218 | 379,022 | – |

Inference is performed at a fraction of a second for all the approaches.

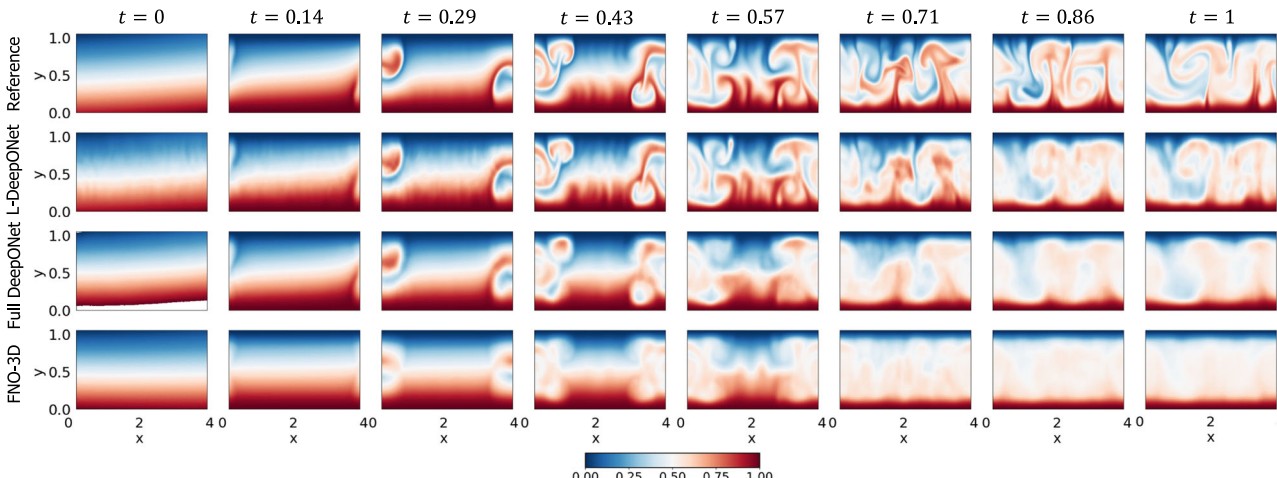

**Fig. 4 | Rayleigh-Bénard convective flow: results of the temperature field of a representative sample for all neural operators.** The results of the L-DeepONet model consider the latent dimension, $d = 100$. The neural operator is trained to approximate the growth of the evolution of the temperature field from a realization of the initial temperature field for seven time steps.

equations can be expressed as

$$\begin{cases} \frac{\mathrm{D}\boldsymbol{V}}{\mathrm{D}t} = -f\boldsymbol{k} \times \boldsymbol{V} - g\nabla h + \nu\nabla^2\boldsymbol{V}, \\ \frac{\mathrm{D}h}{\mathrm{D}t} = -h\nabla \cdot \boldsymbol{V} + \nu\nabla^2 h, \quad \boldsymbol{x} \in \Omega, t \in [0,1], \end{cases} \quad (7)$$

where $\Omega = (\lambda, \phi)$ represents a spherical domain where $\lambda, \phi$ are the longitude and latitude respectively ranging from $[-\pi, \pi]$, $\boldsymbol{V} = \boldsymbol{i}u + \boldsymbol{j}v$ is the velocity vector tangent to the spherical surface ($\boldsymbol{i}$ and $\boldsymbol{j}$ are the unit vectors in the eastward and northward directions respectively and $u, v$ the velocity components), and $h$ is the height field which represents the thickness of the fluid layer. Moreover, $f = 2\Xi \sin\phi$ is the Coriolis parameter, where $\Xi$ is the Earth's angular velocity, $g$ is the gravitational acceleration and $\nu$ is the diffusion coefficient.

As an initial condition, we consider a zonal flow which represents a typical mid-latitude tropospheric jet. The initial velocity component $u$ is expressed as a function of the latitude $\phi$ as

$$u(\phi, t = 0) = \begin{cases} 0 & \text{for} \quad \phi \le \phi_0, \\ \frac{u_{\max}}{n} \exp\left[\frac{1}{(\phi - \phi_0)(\phi - \phi_1)}\right] & \text{for} \quad \phi_0 < \phi < \phi_1, \\ 0 & \text{for} \quad \phi \ge \phi_1, \end{cases} \quad (8)$$

where $u_{\max}$ is the maximum zonal velocity, $\phi_0$, and $\phi_1$ represent the latitude in the southern and northern boundary of the jet in radians, respectively, and $n = \exp[-4/(\phi_1 - \phi_0)^2]$ is a non-dimensional parameter that sets the value $u_{\max}$ at the jet's mid-point. A small unbalanced perturbation is added to the height field to induce the development of barotropic instability. The localized Gaussian perturbation is described as

$$h'(\lambda, \phi, t = 0) = \hat{h}\cos(\phi)\exp[-(\lambda/\alpha)^2]\exp[-(\phi_2 - \phi)/\beta]^2, \quad (9)$$

where $-\pi < \lambda < \pi$ and $\hat{h}, \phi_2, \alpha, \beta$ are parameters that control the location and shape of the perturbation. We consider $\alpha, \beta$ to be random variables with $\alpha \sim U[0.\bar{1}, 0.5]$ and $\beta \sim U[0.0\bar{3}, 0.2]$ so that the input Gaussian perturbation is random. The localized perturbation is added to the initial height field, which forms the final initial condition $h(\lambda, \phi, t = 0)$ (see Supplementary Section on Data Generation). The objective is to approximate the operator $\mathscr{G}: h(\lambda, \phi, t = 0) \mapsto u(\lambda, \phi, t)$. This problem is particularly challenging as the fine mesh required to capture the details of the convective flow both spatially and temporally results in output realizations having millions of dimensions.

Unlike the previous two applications, here the approximated operator learns to map the initial condition of one quantity, $h(\lambda, \phi, t = 0)$, to the evolution of a different quantity, $u(\lambda, \phi, t)$. Given the difference between the input and output quantities of interest (in scale and features), a single encoding of the combined data as in the standard proposed approach (see Fig. 1) is insufficient. Instead, two separate encodings are needed for the input and output data, respectively. While an autoencoder is used to reduce the dimensionality of the output data representing the longitudinal component of the velocity vector $u$, standard principal component analysis (PCA) is performed on the input data due to the small local variations in the initial random height field $h$ which results in a small intrinsic dimensionality.

Results, in terms of MSE, are presented in Fig. 2c, where again we see that the L-DeepONet outperforms the standard approach while changes in the latent dimension do not result in significant differences in the model accuracy. Consistent with the results of the previous application, the training cost of the L-DeepONet is much lower than the full DeepONet (Table 1). We further note that training FNO for this problem (either FNO-2D or FNO-3D) proved computationally prohibitive. For a moderate 3D problem with spatial discretization beyond $64^3$, the latest GPU architectures such as the NVIDIA Ampere GPU do not provide sufficient memory to process a single training sample[49]. Data partitioning across multiple GPUs with distributed memory, model partitioning techniques like pipeline parallelism, and domain decomposition approaches[49] can be implemented to handle high-dimensional tensors within the context of an automatic differentiation framework to compute the gradients/sensitivities of PDEs and thus optimize the network parameters. This advanced implementation is beyond the scope of this work as it proves unnecessary for the studied approach. Consequently, a comparison to the FNO is not shown here. A variant of FNO, the Spherical Fourier neural operator (S-FNO) has been tailored for problems on spherical domains such as shallow water equations[50]. However, S-FNO is primarily focused on mapping the initial condition to the solution at a final timestamp. In contrast, our objective is to learn the mapping from the initial condition to the solution time history, enabling the generation of a sequence of solutions at arbitrary time instants. Hence, we have not provided a comparison between L-DeepONet and S-FNO for this problem. Figure 5, shows the evolution of the L-DeepONet and the full DeepONet compared to the ground truth for a single realization. The L-DeepONet consistently captures the complex nonlinear dynamical features for all time steps, while the full model prediction degrades over time and

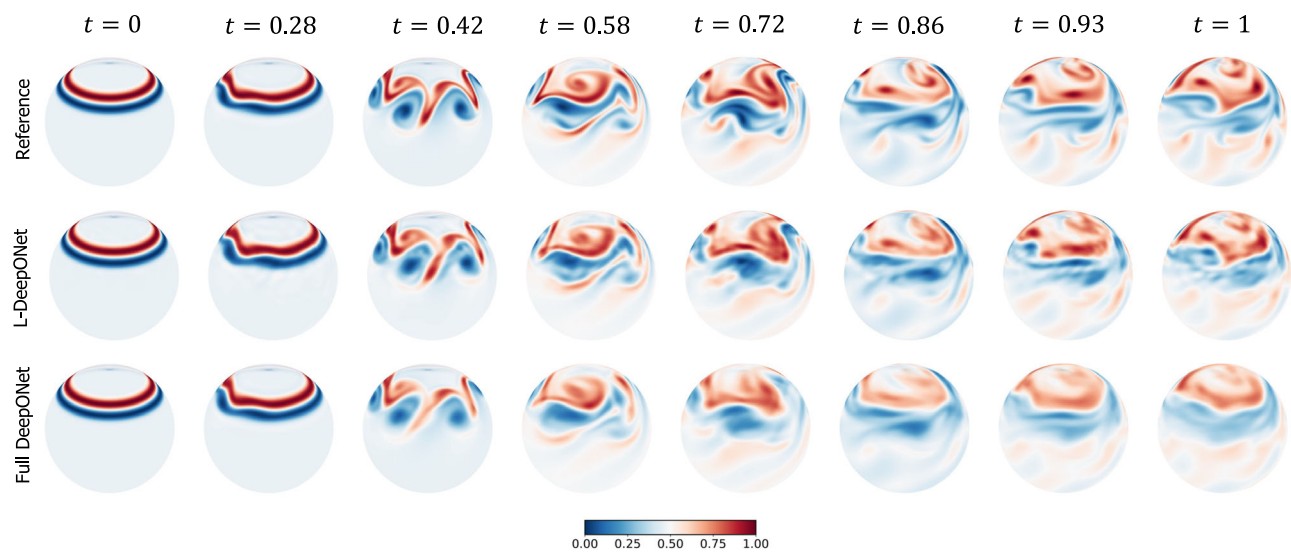

**Fig. 5 | Shallow water equations: results of the evolution of the velocity field through eight time steps for all the operator models considered in this work, for a representative realization of the initial perturbation to the height field.** The results of the L-DeepONet model consider the latent dimension, $d = 81$.

again smoothing the results such that it fails to predict extreme velocity values for each time step that can be crucial, e.g., in weather forecasting.

## Discussion

We have investigated latent DeepONet (L-DeepONet) for learning neural operators on latent spaces for time-dependent PDEs exhibiting highly non-linear features both spatially and temporally and resulting in high-dimensional observations. The L-DeepONet framework leverages autoencoder models to cleverly construct compact representations of the high-dimensional data while a neural operator is trained on the identified latent space for operator regression. Both the advantages and limitations of L-DeepONet are demonstrated in a collection of diverse PDE applications of increasing complexity and data dimensionality. As presented, L-DeepONet provides a powerful tool in SciML and UQ that improves the accuracy and generalizability of neural operators in applications where high-fidelity simulations are considered to exhibit complex dynamical features, e.g., in climate models.

A systematic comparison with standard DeepONet[11] and FNO[29] revealed that L-DeepONet improves the quality of results and it can capture with greater accuracy the evolution of the system represented by a time-dependent PDE. This result is more noticeable as the dimensionality and non-linearity of dynamical features increase (e.g., in complex convective fluid flows). Another advantage is that L-DeepONet training requires less computational resources, as standard DeepONet and FNO are trained on the full-dimensional data and are thus, more computationally demanding and require much larger memory (see Table 1). For all applications, we found that a small latent dimensionality ($d \leq 100$) is sufficient for constructing powerful neural operators, by removing redundant features that can hinder the network optimization and thus its predictive accuracy. Furthermore, L-DeepONet can alleviate the computational demand and thus enable tasks that require the computation of kernel matrices, e.g., used in transfer learning for comparing the statistical distance between data distributions[24].

Despite the advantages of learning operators in latent spaces, there are certain limitations that warrant discussion. L-DeepONet trains DR models to identify suitable latent representations for the combined input and output data. However, as shown in the final application, in cases where the approximated mapping involves heterogeneous quantities, two independent DR models need to be

constructed. While in this work we found that simple MLAE models result in the smallest L-DeepONet predictive error, a preliminary study regarding the suitability of the DR approach needs to be performed for all quantities of interest. Another disadvantage is that the L-DeepONet as formulated is unable to interpolate in the spatial dimensions. The current L-DeepONet consists of a modified trunk net where the time component has been preserved while the spatial dimensions have been convolved. Thus, L-DeepONet can be used for a time but not for space interpolation/extrapolation. Finally, L-DeepONet cannot be readily employed in a physics-informed learning manner since the governing equations are not known in the latent space and therefore cannot be directly imposed. These limitations motivate future studies that continue to assist researchers in the process of constructing accurate and generalizable surrogate models for complex PDE problems prevalent in physics and engineering.

In the context of constructing accurate and generalizable surrogate models, the authors in[51] demonstrate that a slightly modified DeepONet training can achieve an order higher accuracy than its vanilla counterpart[11]. This improvement is realized through the adoption of a two-step training strategy, where the trunk network is initially trained, followed by sequential training of the branch network. The mechanism involves decomposing the entire complex non-convex training task into two subtasks, and the introduction of the Gram-Schmidt orthonormalization process with QR decomposition enhances the stability and generalization capabilities of the model. To demonstrate the effectiveness of the method proposed in[51], we obtained the results for brittle fracture in a plate loaded in shear, shown in Table 2. Furthermore, our observations reveal that substituting QR decomposition with Singular Value Decomposition (SVD) contributes to an enhanced accuracy of the model. The results indicate that the modified training framework successfully mitigates overfitting issues. However, it is worth noting that, in its present form, the framework faces limitations in handling mini-batching of the training dataset.

## Methods
### Problem statement
Neural operators learn nonlinear mappings between infinite dimensional functional spaces on bounded domains and provide a unique simulation framework for real-time inference of complex parametric PDEs. Let $\Omega \subset \mathbb{R}^D$ be a bounded open set and $\mathcal{X} = \mathcal{X}(\Omega; \mathbb{R}^{d_x})$ and $\mathcal{Y} = \mathcal{Y}(\Omega; \mathbb{R}^{d_y})$ two separable Banach spaces. Furthermore, assume that $\mathscr{G} : \mathcal{X} \to \mathcal{Y}$ is a non-linear map arising from the solution of a time-

dependent PDE. The objective is to approximate the nonlinear operator via the following parametric mapping

$$\mathscr{G} : \mathscr{X} \times \Theta \to \mathscr{Y} \quad \text{or,} \quad \mathscr{G}_\theta : \mathscr{X} \to \mathscr{Y}, \theta \in \Theta \tag{10}$$

where $\Theta$ is a finite-dimensional parameter space. In this standard setting, the optimal parameters $\theta^*$ are learned through training the neural operator (e.g., via DeepONet, FNO) with a set of labeled observations $\{\mathbf{x}_j, \mathbf{y}_j\}_{j=1}^N$ generated on a discretized domain $\Omega_m = \{x_1, ..., x_m\} \subset \Omega$ where $\{x_j\}_{j=1}^m$ represent the sensor locations, thus $\mathbf{x}_{j|\Omega_m} \in \mathbb{R}^{D_x}$ and $\mathbf{y}_{j|\Omega_m} \in \mathbb{R}^{D_y}$ where $D_x = d_x \times m$ and $D_y = d_y \times m$. Representing the domain discretization with a single parameter $m$, corresponds to the simplistic case where mesh points are equispaced. However, the training data of neural operators are not restricted to equispaced meshes. For example, for a time-dependent PDE with two spatial and one temporal dimension with discretizations $m_s, m_t$ respectively, the total output dimensionality is computed as $D_y = m_s^{d_x} \times m_t$.

## Approximating nonlinear operators on latent spaces via L-DeepONet

In physics and engineering, we often consider high-fidelity time-dependent PDEs generating very high-dimensional input/output data with complex dynamical features. To address the issue of high dimensionality and improve the predictive accuracy we employ L-DeepONet which allows the training of DeepONet on latent spaces. The approach involves two main steps: (1) the nonlinear DR of both input and output data $\{\mathbf{x}_j, \mathbf{y}_j\}_{j=1}^N$ via a suitable and invertible DR technique, (2) learning of a DeepONet model on a latent space and inverse transformation of predicted samples back to the original space. This process is defined as

$$\begin{aligned} \mathscr{I}_{\theta_{\text{encoder}}} &: \{\mathbf{x}, \mathbf{y}\} \mapsto \{\mathbf{x}^r, \mathbf{y}^r\} \\ \mathscr{G}_\theta &: \mathbf{x}^r \mapsto \mathbf{y}^r \\ \mathscr{I}_{\theta_{\text{decoder}}} &: \mathbf{y}^r \mapsto \mathbf{y}^{\text{rec}} \end{aligned} \tag{11}$$

**Table 2 | Comparative computational accuracy of vanilla DeepONet training and modified DeepONet training[51] for brittle fracture in a plate loaded in shear**

| Training approach | Testing error, when trained | |
|---|---|---|
| | $N_{\text{train}} = 230$ | $N_{\text{train}} = 90$ |
| Vanilla DeepONet | $9 \cdot 10^{-4}$ | $6 \cdot 10^{-3}$† |
| DeepONet with QR decomposition[51] | $2.12 \cdot 10^{-4}$ | $3.77 \cdot 10^{-4}$ |
| DeepONet with SVD | $7.1 \cdot 10^{-5}$ | $8.22 \cdot 10^{-5}$ |

† : Over-fitting observed.

where $\mathscr{I}_{\theta_{\text{encoder}}}, \mathscr{I}_{\theta_{\text{decoder}}}$ are the two parts of a DR method, $r$ corresponds to data on the reduced space, $\mathscr{G}_\theta$ is the approximated latent operator and $\theta$ its trainable parameters. While the encoder $\mathscr{I}_{\theta_{\text{encoder}}}$ is used to project high-dimensional data onto the latent space, the decoder $\mathscr{I}_{\theta_{\text{decoder}}}$ is employed during the training of DeepONet to project predicted samples back to original space and evaluate its accuracy on the full-dimensional data $\{\mathbf{x}_j, \mathbf{y}_j\}_{j=1}^N$. Once trained, L-DeepONet can be used for real-time inference at no cost. We note that the term 'L-DeepONet' refers to the trained DeepONet model together with the pre-trained encoder and decoder parts of the autoencoder which are required to perform inference in unseen samples (see Fig. 1). Next, the distinct parts of the L-DeepONet framework are elucidated in detail.

## Learning latent representations

The first objective is to identify a latent representation for the high-dimensional input/output PDE data. Compressing the data to a reduced representation will not only allow us to accelerate the DeepONet training but, as shown above, improves predictive performance and robustness. To this end, we employ autoencoders `due to their flexibility in the choice of the model architecture and the inherent inverse mapping capability. We note that the proposed framework allows for the adoption of any suitable linear or nonlinear DR method provided the existence of an inverse mapping. In this work, the objective is to demonstrate that DR enhances the accuracy of neural operators rather than establishing which DR method is the most advantageous. The latter depends on various factors including accuracy, generalizability, and computational cost. For our demonstrations, we apply AEs that we found to perform comparably or better than PCA across our diverse set of PDEs through systematic study (see Table 3 and Supplementary Fig. S7). However, the choice of DR approach can be problem and resource-dependent so, although AEs generally outperform PCA, PCA is found to be a viable approach for many problems and under certain conditions.

We train unsupervised autoencoder model $\mathscr{I}_{\theta_{\text{ae}}}$ and perform hyperparameter tuning to identify the optimal latent dimensionality $d$, where $d \ll D_x, D_y$. Assume a time-dependent PDE, where $d_x$ corresponds to the dimensionality of the input space and $m_s, m_t$ the spatial and temporal discretizations of the generated data. In order to feed the autoencoder model with image-like data, the PDE outputs are reshaped into distinct snapshots, i.e., $\{\hat{\mathbf{y}}_i\}_{i=1}^{N \times m_t}$. Finally, input and output data are concatenated into a single dataset $\{\mathbf{z}_i\}_{i=1}^{N(1+m_t)}$. The two parts of the autoencoder model, which are trained concurrently, are expressed as

$$\begin{aligned} \mathscr{I}_{\theta_{\text{encoder}}} &: \{\mathbf{x}, \hat{\mathbf{y}}\} \equiv \mathbf{z} \mapsto \{\mathbf{x}^r, \mathbf{y}^r\} \equiv \mathbf{z}^r, \\ \mathscr{I}_{\theta_{\text{decoder}}} &: \{\mathbf{x}^r, \mathbf{y}^r\} \equiv \mathbf{z}^r \mapsto \{\tilde{\mathbf{x}}, \tilde{\mathbf{y}}\} \equiv \tilde{\mathbf{z}}, \end{aligned} \tag{12}$$

where $\{\mathbf{x}_i^r\}_{i=1}^N \in \mathbb{R}^d$, $\{\mathbf{y}_i^r\}_{i=1}^{N \times m_t} \in \mathbb{R}^d$ and $\{\mathbf{z}_i^r\}_{i=1}^{N(1+m_t)} \in \mathbb{R}^d$. The trainable parameters of the encoder and decoder are represented with $\theta_{\text{encoder}}$

**Table 3 | Comparison of the accuracy of the L-DeepONet for two different dimensionality reduction techniques; namely, the multi-layer autoencoders (MLAE) and principal component analysis (PCA), and $d$ denotes the size of the latent space**

| Application | $d$ | with MLAE | with PCA |
|---|---|---|---|
| Brittle material fracture | 9 | $3.33 \cdot 10^{-4} \pm 4.99 \cdot 10^{-5}$ | $2.71 \cdot 10^{-3} \pm 6.62 \cdot 10^{-6}$ |
| | 64 | $2.02 \cdot 10^{-4} \pm 1.88 \cdot 10^{-5}$ | $3.13 \cdot 10^{-4} \pm 4.62 \cdot 10^{-6}$ |
| Rayleigh-Bénard fluid flow | 25 | $4.10 \cdot 10^{-3} \pm 8.05 \cdot 10^{-5}$ | $3.90 \cdot 10^{-3} \pm 4.73 \cdot 10^{-5}$ |
| | 100 | $3.55 \cdot 10^{-3} \pm 1.46 \cdot 10^{-4}$ | $3.76 \cdot 10^{-3} \pm 4.86 \cdot 10^{-5}$ |
| Shallow water equation | 25 | $2.30 \cdot 10^{-4} \pm 1.50 \cdot 10^{-5}$ | $7.98 \cdot 10^{-4} \pm 8.01 \cdot 10^{-7}$ |
| | 81 | $2.23 \cdot 10^{-4} \pm 1.83 \cdot 10^{-5}$ | $4.18 \cdot 10^{-4} \pm 4.67 \cdot 10^{-6}$ |

Results for both the maximum and minimum $d$ values tested for each application are provided. To evaluate the performance of L-DeepONet, we compute the mean square error of predictions, and we report the mean and standard deviation of this metric based on five independent training trials.

and $\theta_{\text{decoder}}$ respectively. The optimal set of the autoencoder parameters $\theta_{\text{ae}} = \{\theta_{\text{encoder}}, \theta_{\text{decoder}}\}$ are obtained via the minimization of the loss function

$$\mathcal{L}_{\text{ae}} = \min_{\theta_{\text{ae}}} \| \mathbf{z} - \tilde{\mathbf{z}} \|_2^2, \tag{13}$$

where $\| \cdot \|_2$ denotes the standard Euclidean norm and $\tilde{\mathbf{z}} \equiv \{\tilde{\mathbf{x}}, \tilde{\mathbf{y}}\}$ denotes the reconstructed dataset of combined input and output data. From a preliminary study, which is not shown here for the sake of brevity, we investigated three AE models, simple autoencoders (vanilla-AE) with a single hidden layer, multi-layer autoencoders (MLAE), with multiple hidden layers and convolutional autoencoders (CAE) which convolve data through convolutional layers. We found that MLAE performs best, even with a small number of hidden layers (e.g., 3). Furthermore, the use of alternative AE models which are primarily used as generative models, such as variational autoencoders (VAE)[52] or Wasserstein autoencoders (WAE)[53], resulted in significantly worse L-DeepONet performance. Although such models resulted in good reconstruction accuracy and thus can be used to reduce the data dimensionality and generate synthetic yet realistic samples, we found that the obtained submanifold is not well-suited for training the neural operator, as it may result in the reduction of data variability or even representation collapse.

### Training neural operator on latent space (L-DeepONet)

Once the autoencoder model is trained and the reduced data $\{\mathbf{x}^r, \mathbf{y}^r\}$ are generated, we aim to approximate the latent representation mapping with an unstacked DeepONet $\mathcal{G}_\theta$, where $\theta$ are the trainable model parameters. As shown in Fig. 1, the unstacked DeepONet consists of two concurrent DNNs, a branch net which encodes the inputs realizations $\mathbf{x}^r \in \mathbb{R}^d$ (in this case the reduced input data) evaluated at the reduced spatial locations $\{x_1, x_2, \ldots, x_d\}$. On the other hand, the trunk net takes as input the temporal coordinates $\zeta = \{t_i\}_{i=1}^{m_t}$ at which the PDE output is evaluated. The solution operator for an input realization, $\mathbf{x}_1$, can be expressed as:

$$\mathcal{G}_\theta(\mathbf{x}_1^r)(\zeta) = \sum_{i=1}^{p} b_i \cdot tr_i = \sum_{i=1}^{p} b_i(\mathbf{x}_1^r(x_1), \mathbf{x}_1^r(x_2), \ldots, \mathbf{x}_1^r(x_d)) \cdot tr_i(\zeta), \tag{14}$$

where $[b_1, b_2, \ldots, b_p]^T$ is the output vector of the branch net, $[tr_1, tr_2, \ldots, tr_p]^T$ the output vector of the trunk net and $p$ denotes a hyperparameter that controls the size of the final hidden layer of both the branch and trunk net. The trainable parameters of the DeepONet, represented by $\theta$ in Eq. (14), are obtained by minimizing a loss function, which is expressed as:

$$\mathcal{L}(\theta) = \mathcal{L}_r(\theta) + \mathcal{L}_i(\theta),$$
$$\mathcal{L}_r(\theta) = \min_\theta \| \mathbf{y}^r - \tilde{\mathbf{y}}^r \|_2^2, \tag{15}$$

where $\mathcal{L}_r(\theta)$, $\mathcal{L}_i(\theta)$ denote the residual loss and the initial condition loss respectively, $\mathbf{y}^r$ the reference reduced outputs and $\tilde{\mathbf{y}}^r$ the predicted reduced outputs. In this work, we only consider the standard regression loss $\mathcal{L}_r(\theta)$, however, additional loss terms can be added to the loss function. The branch and trunk networks can be modeled with any specific architecture. Here we consider a CNN for the branch net architecture and a feed-forward neural network (FNN) for the trunk net to take advantage of the low dimensions of the evaluation points, $\zeta$. To feed the branch net of L-DeepONet the reduced output data are reshaped to $\mathbb{R}^{\sqrt{d} \times \sqrt{d}}$, thus it is advised to choose square latent dimensionality values. Once the optimal parameters $\theta$ are obtained, the trained model can be used to predict the reduced output for novel realizations of the input $\mathbf{x} \in \mathbb{R}^{D_x}$. Finally, the predicted data are used as inputs to the pre-trained decoder $\mathcal{I}_{\theta_{\text{decoder}}}$, to transform results back to the original space and obtain the approximated full-dimensional output $\mathbf{y}^{\text{rec}} \in \mathbb{R}^{D_y}$. We note that the training cost of L-DeepONet is

**Table 4 | Comparison of the number of trainable parameters for all the neural operators across all considered applications**

| Application | $d$ | L-DeepONet | Full DeepONet | FNO-3D |
|---|---|---|---|---|
| Brittle material fracture | 64 | 76,301 | 144,852 | 6,558,357 |
| Rayleigh-Bénard fluid flow | 100 | 558,258 | 560,128 | 6,558,357 |
| Shallow water equation | 81 | 275,745 | 327,872 | 6,558,357 |

significantly lower compared to the standard model, due to the smaller size of the network and the reduced total number of its trainable parameters (see Table 4).

### Error metric

To assess the performance of L-DeepONet we consider the MSE evaluated on a set of $N_{\text{test}}$ test realizations

$$\text{MSE} = \frac{1}{N_{\text{test}}} \sum_{i=1}^{N_{\text{test}}} (\mathbf{y}_i - \mathbf{y}_i^{\text{rec}})^2, \tag{16}$$

where $\mathbf{y} \in \mathbb{R}^{D_y}$ is the reference and $\mathbf{y}^{\text{rec}} \in \mathbb{R}^{D_y}$ the predicted output respectively. More details on how this framework is implemented for different PDE systems of varying complexity can be found in the Results section. Information regarding the choice of neural network architectures and generation of training data are provided in the Supplementary Tables S2 and S3 as well as Supplementary Section of Data Generation.

### Reporting summary

Further information on research design is available in the Nature Portfolio Reporting Summary linked to this article.

## Data availability

All data needed to evaluate the conclusions in the paper are presented in the paper and/or the Supplementary Materials.

## Code availability

All code accompanying this manuscript is publicly available[54].

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

## Acknowledgements

The authors would like to acknowledge the following funding sources: K.K. and M.D.S.: U.S. Department of Energy, Office of Science, Office of Advanced Scientific Computing Research grant under Award Number DE-SC0020428. S.G. & G.E.K.: U.S. Department of Energy project PhILMs under Award Number DE-SC0019453 and the OSD/AFOSR Multidisciplinary Research Program of the University Research Initiative (MURI) grant FA9550-20-1-0358. Additionally, the authors would like to acknowledge computing support provided by the Advanced Research Computing at Hopkins (ARCH) core facility at Johns Hopkins University and the Rockfish cluster and the computational resources and services at the Center for Computation and Visualization (CCV), Brown University where all experiments were carried out.

## Author contributions

Conceptualization: K.K., S.G., G.E.K., M.D.S., Investigation: K.K., S.G., Visualization: K.K., S.G., Supervision: G.E.K., M.D.S., Writing—original draft: K.K., S.G., Writing—review and editing: K.K., S.G., G.E.K., M.D.S.

## Competing interests

George Karniadakis has financial interests in the companies Anailytica and PredictiveIQ. The rest of the authors declare no competing interests.
