## [Peer Review File · Nature Communications]

REVIEWER COMMENTS

Reviewer #2 (Remarks to the Author):

The work proposes latent DeepONet, a method that uses an encoder to map the input vector to a latent space and uses DeepONet architecture and then a decoder. I have found serious drawbacks in the paper:

1) Wrong statement on discretization invariance: The paper opens with motivations on discretization invariance (in the abstract) and mesh invariance (in the introduction). However, the proposed method is neither discretization invariant nor mesh invariant. The proposed model does not take a function as an input but rather a finite-dimensional vector. The input is the evaluation of a function on pre-defined collocation points. The proposed method is no longer applicable if the resolution changes. Moreover, the authors seem to require point evaluations of the function on pre-specified points. The authors need to clarify how this approach is applicable to PDE problems where the observation points can be arbitrary. As the paper suggests, discretization invariance is desirable when dealing with PDE problems. This property is already satisfied by existing neural operators. So it is unclear why this new method is superior to existing methods.

2) Lack of novelty: The idea of using lower dimensional latent space for operator learning is not novel. In fact, prior work on using an encoder and decoder in operator learning is based on mapping the input function to a latent function (e.g., UNO or U-shaped Neural Operators). However, instead, the authors suggest mapping to a d -dimensional vector in this paper. One drawback of this approach is that, as we increase the input resolution, the latent space does not capture any further information, which is an undesirable property. On the other hand, the previous work (UNO) does not suffer from this drawback since it maps to a latent function. Moreover, one instantiation of UNO is a latent space with a d -dimensional vector, proposed in this paper. So the paper lacks any novelty in model design and superior designs have been proposed previously.

Md Ashiqur Rahman, Zachary E. Ross, Kamyar Azizzadenesheli, U-NO: U-shaped Neural Operators.

3) Wrong statement about FNO: In equation 20, the authors state that FNO computes the Fourier transform and that FNO approximates the Fourier transform in equation 21, using mesh evaluation. This approximation of the integral does not need an equispaced mesh, and Fourier-transform implementations on general meshes exist. Therefore, the authors need to correct their statement, "FNO employs evaluations restricted to an equispaced mesh to discretize both the input and output spaces, where the mesh and the domain must be the same."

4) Lack of careful experimental evaluations: The authors do not show all the details of the experiments, and I am not confident that they have fairly compared with other methods. For instance, the authors claim, “While L-DeepONet results in prediction fields almost identical to the reference, the predictions of the standard models deviate from the ground truth both inside and around the propagated crack.” But why is this happening? It is known that FNO and other neural operators are universal approximators in function spaces. So the authors need to explain why they cannot fit the ground truth well. The authors also do not provide any insight into the number of parameters each model has and the hyperparameter sweeps that were done for the set of models. This holds for all their experiments. The authors must also compare their method with UNO since that is a more appropriate counterpart (vs. FNO) with a reduced latent dimension. For the shallow water equations (SWE), they need to compare against the spherical FNO that has been proposed to handle spherical geometry and attains strong performance.

Boris Bonev, Thorsten Kurth, Christian Hundt, Jaideep Pathak, Maximilian Baust, Karthik Kashinath, Anima Anandkumar "Spherical Fourier Neural Operators: Learning Stable Dynamics on the Sphere" ICML 2023.

Reviewer #4 (Remarks to the Author):

The dynamics of physical systems governed by nonlinear partial differential equations (PDEs) are high-dimensional and multiscale. To predict the complex dynamics has been a challenging problem. Traditional methods often come with great computational expenses and are thus quite limited. In the past decade or so, a sparse optimization approach to finding the underlying PDEs was studied but its applicability is confined to systems with a simple equation structure. In recent years, neural operators stood out as a powerful paradigm of constructing discretization-invariant emulators for PDEs describing physics-based and engineering systems by approximating mappings between infinite-dimensional Banach spaces. The training of the neural operators relies on labeled observational data that are typically generated by high-fidelity simulators. For large systems that involve multiple temporal and spatial scales, generating quality training data becomes challenging.

In this paper, the authors proposed a deep operator network (DeepONet) based approach to this problem. The DeepONet architecture is applied on a low-dimensional latent space that encodes the target dynamics to learn efficient and accurate approximators of the underlying operators. While the idea of training neural operators on latent spaces using DeepONet and autoencoders was

originally proposed in another work in which one of the senior authors of the present manuscript was involved, that work focused on the specific problem of learning two-phase microstructure evolution. The present manuscript addresses the general questions of the predictive accuracy and generalizability of DeepONets trained on latent spaces as compared with conventional neural operators. In addition to carrying out a systematic comparison study, the authors applied Latent-DeepONet to a number of complex physical systems including brittle fracture of materials and complex convective and atmospheric flows, with the finding that Latent-DeepONet consistently outperforms the standard approach in terms of accuracy and computational time.

Overall, the work demonstrated that neural operator learning can be done and is powerful on scales that were previously deemed infeasible. In particular, solving the problem of constructing operators to predict large-scale atmospheric flows with millions of degrees of freedom is impressive, which can be used to improve weather and climate forecasts. The Latent-DeepONet framework is general with broad applications. The paper is very well written. I recommend it highly for Nature Communications.

**RESPONSE TO THE REVIEWERS OF NCOMMS-23-22135-T:
LEARNING HIGH-DIMENSIONAL NONLINEAR OPERATORS IN
LATENT SPACES FOR REAL-TIME PREDICTIONS OF COMPLEX
DYNAMICS IN SCIENCE AND ENGINEERING**

REVIEWER 2

The work proposes latent DeepONet, a method that uses an encoder to map the input vector to a latent space and uses DeepONet architecture and then a decoder. I have found serious drawbacks in the paper:

We genuinely appreciate the diligence with which the reviewer has reviewed our manuscript. We have systemically addressed all the comments. The revisions in the updated manuscript have been highlighted in red color.

- (1) *Wrong statement on discretization invariance: The paper opens with motivations on discretization invariance (in the abstract) and mesh invariance (in the introduction). However, the proposed method is neither discretization invariant nor mesh invariant. The proposed model does not take a function as an input but rather a finite-dimensional vector. The input is the evaluation of a function on pre-defined collocation points. The proposed method is no longer applicable if the resolution changes. Moreover, the authors seem to require point evaluations of the function on pre-specified points. The authors need to clarify how this approach is applicable to PDE problems where the observation points can be arbitrary. As the paper suggests, discretization invariance is desirable when dealing with PDE problems. This property is already satisfied by existing neural operators. So it is unclear why this new method is superior to existing methods.*

We thank the reviewer for the comment. Following are the responses to the queries:

- Achieving discretization invariance is feasible through the utilization of simple interpolation techniques or PCA based approaches, as shown in [1, 2, 3]. These techniques can be integrated at the onset of the framework. Moreover, Roberto Molinaro in his Ph.D. thesis at ETH suggested a simple approach for training DeepONet as shown in Figure 1 (below) that leads to discretization independence. There is also a rigorous proof in the Appendix of [4] to show how DeepONet can be discretization independent. However, in the current work, we have not pursued this, as our focus is on reducing the model size using the latent space. In the revised version, we have removed the words “discretization invariance” and “mesh invariance” to avoid confusion.
- Regarding the “pre-defined” collocation points, the input space of the autoencoder is designed to take as input multiple realizations of parametrized partial differential equations (PDE) within a fixed discretization. In order to accommodate variations in resolution or data at arbitrary spatio-temporal locations, the utilization of projection methods will be crucial. These methods will allow to create a shared continuous basis onto which the training data can be projected. The projected data can then be used for training of

the DeepONet directly. Alternatively, it facilitates the interpolation of the given training data onto a fixed grid, aligning with the requirements of the AE.

- We acknowledge that, in its current state, the L-DeepONet framework may not be suitable for scenarios involving arbitrary points in the spatial domain. However, we recognize this limitation and propose a solution by incorporating either an interpolation framework on a uniform grid or projection methods as mentioned earlier. This adjustment will enhance the framework’s versatility and extend its applicability to scenarios with varying spatial points. For instance, in the work by Lu et al. [5], an interpolation scheme was utilized for converting the unstructured mesh of a complex geometry into a uniform grid (dFNO+ and gFNO+). The primary objective behind this transformation was to facilitate accurate predictions using FNO [6] which requires a lattice grid. A brief discussion about the projection approach along with the limitations of the current method is added in the introduction section of the revised manuscript.

FIGURE 1. Schematic representation of the DeepONet architecture to make it discretization invariant [7].

- (2) *Lack of novelty: The idea of using lower dimensional latent space for operator learning is not novel. In fact, prior work on using an encoder and decoder in operator learning is based on mapping the input function to a latent function (e.g., UNO or U-shaped Neural Operators). However, instead, the authors suggest mapping to a d -dimensional vector in this paper. One drawback of this approach is that, as we increase the input resolution, the latent space does not capture any further information, which is an undesirable property. On the other hand, the previous work (UNO) does not suffer from this drawback since it maps to a latent function. Moreover, one instantiation of UNO is a latent space with a d -dimensional vector, proposed in this paper. So the paper lacks any novelty in model design and superior designs have been proposed previously.*

Md Ashiqur Rahman, Zachary E. Ross, Kamyar Azizzadenesheli, U-NO: U-shaped Neural Operators.

We respectfully disagree with the reviewer’s assertion that the concept of lower-dimensional latent space training for neural operators, as proposed in our work, aligns in spirit with U-shaped Neural Operators. In the proposed approach (employing an autoencoder architecture), a single latent space (L) is defined, characterized by a nonlinear mapping from the input (X) to this space ($E : X \rightarrow L$), and a corresponding mapping from the latent space to the output space ($D : L \rightarrow Y$). The roles of the encoder and decoder are distinct: the encoder transforms the representation of each sample into a modified representation within the latent space, and the decoder reconstructs outputs solely based on this modified representation. This implies that the network can be disassembled, allowing the separate use of the encoder and decoder components [8].

In contrast, within a U-NO architecture, the output mapping is directly tied to the input space. Instead of a direct $L \rightarrow Y$ mapping, a skip connection is employed, represented as $[X + L] \rightarrow Y$. Although the U-Net framework in U-NO is commonly categorized into encoder and decoder segments, these do not function as traditional (and distinct) encoder and decoder components. In typical encoder-decoder architectures, the sample is mapped onto a well-defined latent space, and the output is computed from this space. However, in U-Net, the input and all intermediate representations are essential to generate the output. Consequently, the encoder and decoder in a U-NO cannot be separated into two independent entities for computing the output.

Therefore, the fundamental distinction lies in the ability to disentangle and use distinct encoder and decoder components separately, a characteristic that differs significantly from the architecture of U-Net. This provides the additional freedom to choose the form of the encoder and decoder operations, making it suitable for adoption with many other nonlinear dimension reduction schemes (including those that are not neural network-based such as Diffusion Maps). In this work, we use a simple multi-layer autoencoder (as outlined in the supplement), but have also compared to non-neural network based dimension reduction scheme (namely PCA). Importantly, the L-DeepONet is not tied to the architecture/design of the encoder/decoder. In the revised manuscript, we have incorporated a concise discussion highlighting the distinctions between our method and U-NO. This discussion can be found in both the introduction sections and the supplementary materials, providing readers with insights into the unique aspects and contributions of our approach.

Another distinction is that the spatial basis for U-NO is discrete unlike the continuous trunk basis of DeepONet. Furthermore, the authors in [9] show that DeepONet can achieve SOTA results with an efficient and slightly modified training approach. To demonstrate the effectiveness of the method proposed in [9], we obtained the results for brittle fracture in a plate loaded in shear, shown in Table 1. The results show that with the modified training framework, we can obtain an order higher accuracy. So, taken together, the two new training techniques proposed in Molinaro and Shin make DeepONet superior to existing architectures for realistic 3D problems. We have added a brief discussion about the modified training framework proposed by [9] in the Discussion section of the revised manuscript.

Regarding the non-significant improvement with the increase in resolution of the

latent space; in any dimensionality reduction approach, a crucial step involves determining the number of components required to capture a substantial percentage of the total variance in the data, defined by a pre-determined threshold. Beyond this point, increasing the number of components does not yield significant improvements in capturing additional information in the latent space. Similarly, within our framework, our goal is to identify the number of components in the latent space that are sufficient for accurately reconstructing the physical space, thereby explaining the observation made by the reviewer.

TABLE 1. Comparative computational accuracy of vanilla DeepONet training and modified DeepONet training [9] for brittle fracture in a plate loaded in shear.

Training Approach	Testing error, when trained	
	$N_{\text{train}} = 230$	$N_{\text{train}} = 90$
Vanilla DeepONet	$9 \cdot 10^{-4}$	$6 \cdot 10^{-3}\dagger$
DeepONet with QR decomposition [9]	$2.12 \cdot 10^{-4}$	$3.77 \cdot 10^{-4}$
DeepONet with SVD	$7.1 \cdot 10^{-5}$	$8.22 \cdot 10^{-5}$

\dagger : Over-fitting observed.

- (3) *Wrong statement about FNO: In equation 20, the authors state that FNO computes the Fourier transform and that FNO approximates the Fourier transform in equation 21, using mesh evaluation. This approximation of the integral does not need an equispaced mesh, and Fourier-transform implementations on general meshes exist. Therefore, the authors need to correct their statement, “FNO employs evaluations restricted to an equispaced mesh to discretize both the input and output spaces, where the mesh and the domain must be the same.”*

We agree with the reviewer that Fourier-transform implementations exist on general meshes. However, it is important to emphasize that our focus in this work is on the principles outlined in the seminal work of FNO [6], where the framework’s derivation specifically considers a uniform discretization. Notably, the authors of FNO highlight that, despite the exceptional efficiency of fast Fourier transforms, they necessitate a uniform discretization. To provide clarity on this aspect, we have incorporated a citation to the seminal work of FNO in the quoted sentence within the revised version of the manuscript.

- (4) *Lack of careful experimental evaluations: The authors do not show all the details of the experiments, and I am not confident that they have fairly compared with other methods. For instance, the authors claim, “While L-DeepONet results in prediction fields almost identical to the reference, the predictions of the standard models deviate from the ground truth both inside and around the propagated crack.” But why is this happening? It is known that FNO and other neural operators are universal approximators in function spaces. So the authors need to explain why they cannot fit the ground truth well. The authors also do not provide any insight into the number of parameters each model has and the hyperparameter sweeps that were done for the set of models. This holds for all their experiments. The authors must also compare their method with UNO since that is a more appropriate counterpart (vs. FNO) with a reduced latent dimension. For the shallow water equations (SWE), they need to compare against the spherical FNO that has been proposed to handle spherical geometry and attains strong performance.*

- The deviation from the ground truth is attributed to the total error – consisting of approximation error, empirical error and the generalization error; the approximation error is typically the smallest of the three.
- In the revised manuscript, we have introduced Table 3 (Table 2 in the response to reviewer document) to provide a comprehensive overview of the number of trainable parameters within each of the investigated frameworks.

TABLE 2. Comparison of the number of trainable parameters for all the neural operators across all considered applications.

Application	d	L-DeepONet	Full DeepONet	FNO-3D
Brittle material fracture	64	76,301	144,852	6,558,357
Rayleigh-Bénard fluid flow	100	558,258	560,128	6,558,357
Shallow water equation	81	275,745	327,872	6,558,357

- To address the question of hyperparameter sweeps, in Figure 2 of the manuscript we have presented the mean square error (MSE) values corresponding to various latent dimensions, specifically $d = [25, 49, 64, 81]$, for both the autoencoder architecture and the Latent-DeepONet to explicitly show the changes in accuracy of the framework as we change the latent dimension size of the AE.
- We agree that the Spherical Fourier neural operator (S-FNO) has been specifically designed for problems on spherical domains such as the shallow water equations, focusing on mapping the initial condition to the solution at a particular time, such as $t = t_1$. In contrast, our manuscript addresses the mapping of the initial condition to the solution time history, generating a sequence of solutions at arbitrary time instants. Consequently, the current S-FNO code is not applicable for solving our problem due to the differences in the nature of the mapping tasks. In the Results section of the revised manuscript, we have acknowledged S-FNO for SWE and explicitly elucidated the reasons behind the unfeasibility of presenting a comparative result.
- Based on our earlier responses, we believe that the U-NO framework and the L-DeepONet framework differ significantly in their design, addressing distinct challenges within the original FNO and DeepONet architectures, respectively. Specifically, the U-NO architecture mirrors the U-Net design, enabling the efficient training of overparameterized models and harnessing the advantages of DNNs. Nevertheless, akin to FNO, U-NO employs convolutional kernel operations to approximate the linear integral operation in the Fourier domain using the fast Fourier transform. To underscore our argument, we have conducted simulations of U-NO across all examples discussed in the paper. The results, including computational time in seconds and accuracy, are presented in Table 3 of the response to the reviewer document. It is crucial to highlight that U-NO encounters the same computational challenges as FNO in the context of high-dimensional PDEs due to the intensive computational demands associated with the Fourier transform. Therefore the Shallow water equation was intractable with the U-NO and the computational time and accuracy could not be reported. We have included a brief discussion on the

U-NO architecture and results presented in Table 3 to the supplementary materials of the revised manuscript.

TABLE 3. Accuracy, computational time in seconds and number of trainable parameters for U-NO across all considered applications.

Application	Accuracy	Time/epoch	# trainable parameters
Brittle material fracture	$0.16 \cdot 10^{-4}$	3,136	1,280,000
Rayleigh-Bénard fluid flow	$2.22 \cdot 10^{-4}$	7,763	70,993,525
Shallow water equation	--	--	1,135,887,697

The dynamics of physical systems governed by nonlinear partial differential equations (PDEs) are high-dimensional and multiscale. To predict the complex dynamics has been a challenging problem. Traditional methods often come with great computational expenses and are thus quite limited. In the past decade or so, a sparse optimization approach to finding the underlying PDEs was studied but its applicability is confined to systems with a simple equation structure. In recent years, neural operators stood out as a powerful paradigm of constructing discretization-invariant emulators for PDEs describing physics-based and engineering systems by approximating mappings between infinite-dimensional Banach spaces. The training of the neural operators relies on labeled observational data that are typically generated by high-fidelity simulators. For large systems that involve multiple temporal and spatial scales, generating quality training data becomes challenging.

In this paper, the authors proposed a deep operator network (DeepONet) based approach to this problem. The DeepONet architecture is applied on a low-dimensional latent space that encodes the target dynamics to learn efficient and accurate approximators of the underlying operators. While the idea of training neural operators on latent spaces using DeepONet and autoencoders was originally proposed in another work in which one of the senior authors of the present manuscript was involved, that work focused on the specific problem of learning two-phase microstructure evolution. The present manuscript addresses the general questions of the predictive accuracy and generalizability of DeepONets trained on latent spaces as compared with conventional neural operators. In addition to carrying out a systematic comparison study, the authors applied Latent-DeepONet to a number of complex physical systems including brittle fracture of materials and complex convective and atmospheric flows, with the finding that Latent-DeepONet consistently outperforms the standard approach in terms of accuracy and computational time.

Overall, the work demonstrated that neural operator learning can be done and is powerful on scales that were previously deemed infeasible. In particular, solving the problem of constructing operators to predict large-scale atmospheric flows with millions of degrees of freedom is impressive, which can be used to improve weather and climate forecasts. The Latent-DeepONet framework is general with broad applications. The paper is very well written. I recommend it highly for Nature Communications.

We thank the review for the appreciation and positive feedback.

REFERENCES

- [1] Yong Zheng Ong, Zuwei Shen, and Haizhao Yang. Integral autoencoder network for discretization-invariant learning. *The Journal of Machine Learning Research*, 23(1):12996–13040, 2022.
- [2] Maarten V de Hoop, Daniel Zhengyu Huang, Elizabeth Qian, and Andrew M Stuart. The cost-accuracy trade-off in operator learning with neural networks. *arXiv preprint arXiv:2203.13181*, 2022.
- [3] Kaushik Bhattacharya, Bamdad Hosseini, Nikola B Kovachki, and Andrew M Stuart. Model reduction and neural networks for parametric pdes. *The SMAI journal of computational mathematics*, 7:121–157, 2021.
- [4] Francesca Bartolucci, Emmanuel de Bezenac, Bogdan Raonic, Roberto Molinaro, Siddhartha Mishra, and Rima Alaifari. Representation equivalent neural operators: a framework for alias-free operator learning. In *Thirty-seventh Conference on Neural Information Processing Systems*, 2023.
- [5] Lu Lu, Xuhui Meng, Shengze Cai, Zhiping Mao, Somdatta Goswami, Zhongqiang Zhang, and George Em Karniadakis. A comprehensive and fair comparison of two neural operators (with practical extensions) based on FAIR data. *Computer Methods in Applied Mechanics and Engineering*, 393:114778, 2022.
- [6] Zongyi Li, Nikola Kovachki, Kamyar Azizzadenesheli, Burigede Liu, Kaushik Bhattacharya, Andrew Stuart, and Anima Anandkumar. Fourier neural operator for parametric partial differential equations. *arXiv preprint arXiv:2010.08895*, 2020.
- [7] Roberto Molinaro, Yunan Yang, Björn Engquist, and Siddhartha Mishra. Neural inverse operators for solving pde inverse problems. *arXiv preprint arXiv:2301.11167*, 2023.
- [8] Thomas Schlegl, Philipp Seeböck, Sebastian M Waldstein, Georg Langs, and Ursula Schmidt-Erfurth. f-AnoGAN: Fast unsupervised anomaly detection with generative adversarial networks. *Medical image analysis*, 54:30–44, 2019.
- [9] Sanghyun Lee and Yeonjong Shin. On the training and generalization of deep operator networks. *arXiv preprint arXiv:2309.01020*, 2023.

REVIEWERS' COMMENTS

Reviewer #4 (Remarks to the Author):

The authors did a good job to revise the paper according to the referee comments. The critical comments of Reviewer #2 have been fully addressed. The present version of the manuscript is suitable for Nature Communications - I recommend it highly.